# MicroRNAs: Small Molecules with Significant Functions, Particularly in the Context of Viral Hepatitis B and C Infection

**DOI:** 10.3390/medicina59010173

**Published:** 2023-01-15

**Authors:** Fayed Megahed, Ashraf Tabll, Shimaa Atta, Ameera Ragheb, Robert Smolic, Ana Petrovic, Martina Smolic

**Affiliations:** 1Nucleic Acid Research Department, Genetic Engineering and Biotechnological Research Institute (GEBRI), City for Scientific Researches and Technological Applications (SRTA-City), Alexandria 21934, Egypt; 2Microbial Biotechnology Department, National Research Centre, Giza 12622, Egypt; 3Egypt Center for Research and Regenerative Medicine (ECRRM), Cairo 11517, Egypt; 4Department of Immunology, Theodor Bilharz Research Institute, Cairo 12411, Egypt; 5Faculty of Dental Medicine and Health Osijek, University of Osijek, Crkvena 21, 31000 Osijek, Croatia

**Keywords:** microRNAs, hepatitis B virus, hepatitis C virus, diagnosis, biomarkers

## Abstract

A MicroRNA (miRNA) is defined as a small molecule of non-coding RNA (ncRNA). Its molecular size is about 20 nucleotides (nt), and it acts on gene expression’s regulation at the post-transcription level through binding to the 3’untranslated regions (UTR), coding sequences, or 5’UTR of the target messenger RNAs (mRNAs), which leads to the suppression or degradation of the mRNA. In recent years, a huge evolution has identified the origin and function of miRNAs, focusing on their important effects in research and clinical applications. For example, microRNAs are key players in HCV infection and have important host cellular factors required for HCV replication and cell growth. Altered expression of miRNAs affects the pathogenicity associated with HCV infection through regulating different signaling pathways that control HCV/immunity interactions, proliferation, and cell death. On the other hand, circulating miRNAs can be used as novel biomarkers and diagnostic tools for HCV pathogenesis and early therapeutic response. Moreover, microRNAs (miRNA) have been involved in hepatitis B virus (HBV) gene expression and advanced antiviral discovery. They regulate HBV/HCV replication and pathogenesis with different pathways involving facilitation, inhibition, activation of the immune system (innate and adaptive), and epigenetic modifications. In this short review, we will discuss how microRNAs can be used as prognostic, diagnostic, and therapeutic tools, especially for chronic hepatitis viruses (HBV and HCV), as well as how they could be used as new biomarkers during infection and advanced treatment.

## 1. Introduction

Twenty years ago, the origin and applications of miRNAs were unknown. Until then, scientists were concise about expressed genes that translated into proteins. The classical central dogma of biology, that DNA is transcribed into RNA which then translates into protein, neglected research about non-protein-coding genomes. Not until 1993 did the significance and functions of miRNAs become clear [1,2].

There are different types of small endogenous RNAs, such as small transfer RNA (tRNA), ribosomal RNA (rRNA), small nucleolar RNA (snoRNA), small interfering RNA (siRNA) and microRNA (miRNA). The biochemistry and functions of miRNA and siRNA are uncharacterized; they are about 19–20 nucleotides (nt) in length with 5′-phosphate and 3′-hydroxyl ends, and are assembled into RISC to silence or inhibit specific gene expression [3,4]. Therefore, these siRNA and microRNA are distinguished according to their origins, as microRNAs are generated from small RNA hairpin precursor double-stranded RNA (60–70 nt), while siRNA is derived from long double-stranded RNA (dsRNA) [3,5]. Mainly, all small RNAs that mediate post-transcriptional modifications by gene silencing through RISC formation have been expressed as siRNA, regardless of their precursor.

## 2. MicroRNAs’ Synthesis and Mechanism of Action

In the genome, the miRNAs are formed from pre-miRNA, which is derived from pri-miRNAs, which are transcribed from the genome by RNA polymerases (II and III) characterized by long primary transcripts that are 5-capped and 3-polyadenylated. By cellular RNase II endonuclease III Drosha and DGCR8/Pasha proteins, Pri-miRNAs are converted into precursor-miRNAs (pre-miRNAs), which are about 60–110 nt in length and are transported from the nucleus to the cytoplasm by an Exportin-5-dependent effect. Pre-miRNA is cleft by the RNase III enzyme Dicer-1 and TRBP/PACT proteins to produce a short double-stranded miRNA duplex in the cytoplasm. Then, the miRNA duplex is unwound by a helicase enzyme into a mature miRNA (20 nt), which is then inserted into a multicomponent complex constituted by Argo family protein members, known as RISC (RNA-induced silencing complex), which leads to translation suppression or mRNA degradation [6].

## 3. Biological Functions of MicroRNAs

Gene regulation is the main job of miRNAs in the human body. They accomplish this by regulating mRNA expression as well as by controlling transcription and translation. There are two pathways of miRNA synthesis: canonical and non-two successive [7,8]. The non-canonical mechanism is when the miRISC complex, which includes the miRNA-guided strand, binds to the target mRNA by means of its 3′UTR through the first 2–7 nucleotides from the 5-end. This is followed by mRNA deadenylation, translation suppression, and mRNA degradation [9,10]. This process occurs with the deep sequence of the miRNA, the first 2–7 nucleotides from the 5′ end, and it is followed by mRNA deadenylation, translation suppression, and, finally, degradation (Figure 1) [5,11]. This is the second way that miRISC complexes and mRNAs interact with each other. It affects about 60% of these interactions, which have sequences that are not entirely complementary. Many biological processes are regulated by this interaction through non-canonical pathways, which could allow a single miRNA to target many mRNAs or a single mRNA to contain multiple binding sites for miRNAs [12,13].

Although miRNAs significantly participate in intercellular signaling, circulating miRNAs (exogenous miRNAs) have been defined as important signaling molecules in extracellular signaling, and are defined as miRNAs that are found inside cells that can migrate outside in body fluids, such as blood, urine, saliva, seminal fluid, breast milk, or other fluids through tissue damage, apoptosis, and necrosis, either via active passage, in microvesicles or exosomes, or through protein bonding [14,15]. A huge amount of scientific progress has been made to accurately find the origin, functions, and clinical applications of miRNA for healthy and disease-carrying patients. The length and intensity of exercise also have effects on the inflammation, angiogenesis, and cardiac muscle contractility of the heart. Previous studies have found that some circulating miRNAs play a role in these processes [16,17].

## 4. The Potential Role of miRNAs as Biomarkers for Diseases

Biomarkers are different types of objectives that indicate the presence or absence of a disease in an individual. For the first time, Lawrie et al. [18] used miRNAs as biomarkers to examine diffused large B-cell lymphoma in the serum of patients [18,19], and ever since then, miRNAs have been utilized as specific biomarkers for different diseases. The ideal biomarkers should be applicable, specific, and easily accessible. These features are found in miRNAs, which could be easily isolated from body fluids and biopsies to act as sensitive and specific biomarkers for disease progression, cancer stages, and therapeutic response. On the other hand, nucleic acid technologies have been established and could be applicable in miRNA assays, as they require a short time period and have a low cost, rather than using the new antibodies for protein biomarkers [20].

Using miRNAs as multi-markers can provide accurate diagnoses as well as targeted therapy and efficient treatment, similarly to protein markers, which are often costly and time-consuming. For example, urinary miRNA is a signature of lupus nephritis that can aid in detecting early renal fibrosis [21]. Moreover, miRNA could be more specific, as a multi-marker approach, in cancer diagnosis and heterogeneous diseases. Until now, 9-miRNA multi-markers have specifically been applied to breast cancer diagnosis [22]. MiRNAs’ role in biomarker research is still considered insufficient due to a lack of reproducibility and discordance reported between different studies for the same tumor diagnosis [23]. To solve this problem, standard methods must be developed, such as sampling, transport, storage, and data analysis.

## 5. MicroRNAs in Liver Diseases

The hepatitis B virus (HBV) is a partially double-stranded DNA virus (3.2 kb) and is non-cytopathic, with a complicated replicative cycle. Numerous reports have evaluated the important roles of specific miRNAs in HBV pathogenesis and different stages of infection. Liver diseases can be classified into chronic liver disease, liver cirrhosis, and hepatocellular carcinoma (HCC), which may occur with multiple viral infections, such as HCV, HDV, HAV, and HEV, along with other diseases associated with environmental factors, nutrition, drug toxicity, or alcohol abuse. Otherwise, miRNAs are considered as specific marks and accurate biomarkers for the etiology and progression of liver diseases (Figure 2) [24,25].

### 5.1. Chronic Hepatitis

Previous studies have reported that miR-122 is a liver-specific miRNA, and is related to chronic liver diseases. MiR122 is up-regulated significantly in blood samples of HBeAg-positive chronic hepatitis patients [26]. Another study reported increasing levels of miR372/373 with increased HBV DNA titer; hence, it is considered a chronic hepatitis marker [27]. Chen et al. [28] have shown that let-7c, miR23b, miR122, and miR150 can be applied as diagnostic tools for occult HBV infection [28].

### 5.2. Liver Cirrhosis

MiR29 has served as an accurate biomarker for liver cirrhosis, fibrosis, and necro-inflammation [29,30]. It prevents liver cirrhosis and fibrosis when it is up-regulated. When this happens, it stops the production of collagen-secreting genes in hepatic stellate cells (HSCs), which are found in the hepatocyte matrix. This happens through the TGF- and NF-kB pathway [31,32]. MiR133a and miR199 are down-regulated during liver fibrosis and cirrhosis [33,34]. On the other hand, miR181b, miR214-5p, miR221, and miR222 are up-regulated during cirrhosis, and can be used as specific markers [35,36].

### 5.3. Hepatocellular Carcinoma (HCC)

HBV X protein (HBx) is a protein that plays a big role in the life cycle of HBV. It also affects the expression of many cellular miRNAs, which helps the hepatocarcinogenesis process. Such miRNAs include MiRs 17–92 cluster (mir-17-5p, miR-18a, miR-19a, miR-19b, miR-20a, and miR-92a-1), and miR21 plays an important role in HBV-related HCC [37]. In addition, Laderio et al. [38] reported the over-expression of miR-96 in HBV-related HCC as compared to non-HBV-related HCC. MiRNAs, including miR-375 and miR-92a, are significantly expressed in HBV-associated HCC cases, which was tested by RT-PCR assay and sequencing [39]. Moreover, miR-145 and miR-199b are down-regulated, while miR-244 and miR155 are up-regulated, in small dysplastic nodules (DNs) and HCC [40]. There are two things that happen when the HBx protein hyper methylates the promoter of miR-122 at CpG islands. First, it stops the PPAR and RXR from binding to the promoter, which causes the down-regulation of miR-122 expression in liver cancer [41]. Moreover, Zhao et al. [42] reported that high-serum miR-324-3p may serve as a diagnostic and prognostic biomarker for HBV-related HCC.

### 5.4. MicroRNAs in HBV and Related Diseases

HBV is a small, enveloped DNA virus from the *Hepadnaviridae* family. It mainly infects hepatocytes and leads to acute and chronic liver diseases, cirrhosis, and liver cancer. Globally, about 2 billion people are infected with HBV, while 350 million are carriers and have chronic infections [43,44]. However, the HBV vaccine is available and efficient. HBV infection remains a challenge and a global health problem. The advancement in the application of miRNAs in HBV infection has led us to understand the molecular biology and pathogenicity of HBV [45,46].

#### 5.4.1. Effects of miRNAs on HBV Infection and Replication

HBV modulates the gene expression of several cellular miRNAs to promote a better environment for its replication and survival [47]. There are many different miRNAs that play a significant role in how HBV spreads during both acute and chronic infections, such as miR-122, which is very high in normal hepatocytes. This miRNA has an effect on liver function and lipid metabolism [48]. In addition, miR-122 acts as a negative regulator of HBV genetic expression through binding to HBV conserved regions [28]. A new report has shown that miR-122 dysregulation can be implicated by HBV X proteins (HBx) [41]. In addition, miR-1 helps with HBV core promoter transcription by lowering the levels of histone deacetylase 4 (HDAC4). This helps nuclear HBx to improve epigenetic changes to HBV cccDNA in order to amplify the HBV genome [49]. Moreover, the let-7 family of miRNAs is negatively regulated by HBx to increase the signal transducer and activator of transcription 3 (STAT3) activities, which enhance cell proliferation, viral replication, and liver carcinogenesis [50]. In addition, miR-501 promotes HBV replication and hepatocarcinogenesis [51]. Lastly, Wang et al. [52] recently reported that serum exosomal hsa_circ_0028861 might influence HCC progression by regulating its targeted microRNAs (miRNAs) and downstream tumor and related signaling pathways, and that this serum marker could be used as a novel diagnostic tool for HBV-derived HCC.

#### 5.4.2. Role of miRNAs in HBV Immunity

MiR-155 has multiple functions in innate immunity, as it regulates the acute inflammatory process. After the pathogens have been recognized by the toll-like receptors, it stimulates the innate immunity via the janus kinase JAK/STAT pathway and diminishes HBx expression [53,54]. On the other hand, miR-181a is up-regulated in hepatoma cell lines and activates HBV replication through suppression of the human leukocyte antigen A (HLA-A)-dependent HBV antigen presentation [45]. Moreover, miR-181a and miR-146 are altered in hepatocytes and activate HBV evasion [55].

#### 5.4.3. Effects of miRNAs in HBV Chronic Infection

Down-regulation of MiR-152 can induce DNA methyltransferase1 (DNMT1) over-expression by the effect of HBx to prevent HBV gene expression and antigen presentation [56]. A previous study reported that miR-1 enhances HBV replication and inhibits cell proliferation, inducing a reverse cancer cell phenotype [57]. In addition, miR-122 binds directly to the HBV polymerase gene to inhibit its expression [36]. Previous studies have reported that miR-125a-5p and miR-199a-3p affect the HBVS region, and that miR-210 interacts with the pre-S1 region [57,58].

#### 5.4.4. Role of miRNAs in HBV-Related Diseases (Cirrhosis and HCC)

Connolly et al. reported that the elevated expression of the miR-17-92 cluster acts as a biomarker in the malignant phenotype [37]. Moreover, Liu et al. [37,59] studied the role of serum miR-122 as a marker of HBV that caused liver injury. MiR-122 is usually at its highest in people who have chronic hepatitis and then cirrhosis, at the point when they go from being carriers of HBV to having full-blown hepatitis and, subsequently, cirrhosis. Thus, the level of miR-122 in the blood could be used as a biomarker to check for HBV, which causes liver damage [37].

### 5.5. MicroRNAs in HCV Infection and Related Diseases

Advanced reports have established that miRNAs have a pivotal role in HCV infection and pathogenesis. Many cellular miRNAs suppress the HCV RNA replicative cycle, while others enhance it [60,61].

#### 5.5.1. MicroRNAs That Inhibit HCV Replication

Through analyzing the expression of different cellular miRNAs in IFN-stimulated cells by microarray technology, Pedersen et al. found that the levels of about 30 miRNAs were changed (up-regulated or down-regulated). They also detected 8 IFN-β-induced miRNAs (miR-1, -30, -128, -196, -296, -351, -431, and -448) which interact with the HCV genome and are over-expressed by miRNA mimics transfection to achieve an antiviral response of IFN-β in Huh7 cells [62]. Another study found that miR-196 diminishes HCV RNA replication in HCV replicon cells [63].

#### 5.5.2. MicroRNAs That Enhance HCV Replication

MiR-122 is most predominant in the liver (about 70% content), playing a major role in fatty acid metabolic regulatory pathways. Its reduced level is linked with liver cancer and regulates HCV replication in hepatocytes [64,65]. Jopling et al. [66] reported the molecular function of miR-122 in HCV replication by using antisense oligonucleotide technology to decrease HCV RNA accumulation and thus discovering two target sites in the HCV genome that are essential for its replication [66,67,68].

#### 5.5.3. Function of MicroRNAs as Regulators for HCV/ INNATE Immunity Interactions

HCV inhibits the IFITM1 gene, an interferon stimulated gene, via up-regulation of miR-130a expression in HCV-infected hepatocytes, so that HCV also alters multiple miRNAs to inhibit the IFNI signaling pathway [69]. On the other hand, miR-122 over-expression suppresses the IFN signaling pathway through altered expression of SOCS3 [70]. Moreover, increased levels of miR-21 inhibit MyD88 and IRAK1 expression in hepatocytes, which consequently suppresses type I IFN effector gene expression and promotes viral replication. However, miR-324-5p and miR-489 are up-regulated with IFN-α, which can modulate HCV-specific miRNA expression in hepatocytes [71].

#### 5.5.4. Effect of MicroRNAs in HCV-Associated Inflammatory Fibrotic Process

Numerous miRNAs have been reported in different types of cancer as oncogenes or tumor suppressor genes. MiR-449a is down-regulated in HCV patients, but not in other chronic liver diseases, as it is up-regulated by YKL40 (an inflammatory marker) through targeting of the NOTCH signaling pathway after an HCV infection [72]. In addition, the increased expression of miR-155 enhances hepatocyte proliferation and carcinogenesis via the Wnt signaling pathway, and the expression levels of miR-21 positively modulate liver fibrosis during HCV infection [23].

#### 5.5.5. Role of MicroRNAs in HCC-Related HCV Infection

Different miRNAs have been associated with the initiation and progression of hepatocellular carcinoma. Despite the defined role of miRNAs in HCC-related HCV infection, it is still unclear whether the specific detection of HCC-related miRNA has a great impact as a prognostic or early diagnostic marker of HCC [73,74]. On the other hand, miRNA expression analysis from HCC-related HCV samples showed that the levels of 10 miRNAs increased, and those of 19 miRNAs decreased [75]. However, the validation of the accuracy of these miRNAs, their predicted targets, and their application as specific biomarkers of HCC in HCV patients requires more studies. Recently, Petkevich et al. [76] evaluated exosomal and non-exosomal miRNAs, such as let-7a-5p, -16-5p, -18a-5p, -21-5p, -22-3p, -34a-5p, -103a-3p, -122-5p, -221-3p, and -222-3p, in plasma and saliva samples of patients with HCV-related liver cirrhosis and primary liver cancer, as well as control samples, to find that the non-exosomal miRNAs (let-7a, miRNA-21-5p, -22-3p, -103a, -122-5p, -221-3p and 222-3p) were normalized to non-exosomal miRNA-16-5p in the plasma of HCC patients. Three microRNAs (miRNA-21-5p, 122-5p, and 221-3p) were found in saliva. Thus, the researchers concluded that miRNA fractions may serve as candidate diagnostic and prognostic markers. For example, Abdelkhalek et al. [77] reported that MiRNA-21 may be a useful, non-invasive tool for diagnosing HCC. Additionally, non-cirrhotic HCV patients have five times the risk of developing HCC if their miRNA -21 level is ≤1.4468.

#### 5.5.6. MicroRNAs in HBV/HCV Therapy

HCV antisense oligonucleotide (SPC3649) is a complementary target of the 5′-end of miR-122 which has been estimated to reduce HCV RNA titers in HCV-infected chimpanzees’ liver and blood samples [78]. In addition, miravirsen (antisense oligonucleotide for miR-122) decreased HCV’s viral load without resistance in chronic HCV genotype 1-infected patients (phase II clinical trial), but had no adverse side effects in HCV-infected patients or chimpanzees [79].

For RNAi-based inhibition of HBV replication, Liu et al. [80] reported that an adenosine deaminase that acts on RNA-1 (ADAR1) involves adenosine-to-inosine RNA editing and microRNA processing. ADAR1 is known to be involved in the replication of various viruses, such as HCV, HDV, and HBV. Liu et al. [80] reported that ADAR1 plays an antiviral role in HBV infection by increasing the level of miRNA-122 in hepatocytes. Another approach for HBV infection considers that miR-29b mimics prevent liver fibrosis [81], and miR-99a mimics suppress tumor growth in HCC [39]. Recently, Chen et al. [82] presented miRNA-30b-5p/MINPP1 as a novel biomarker for HBV-related HCC as an early diagnostic tool and a potential pharmaceutical target for antitumor therapy. Previous research has shown that miRNAs enter the bloodstream via many organs and tissues, such as the brain, heart, endothelial cells, genitourinary system, and mammary glands, in both healthy and diseased individuals. Circulating miRNAs originate from tumor tissues, and, therefore, may be able to assess cancer progression. It has been noted that after tumor resection, oncogenic miRNA levels decrease [83]. Numerous techniques have been carried out to detect, quantify, and analyze miRNAs, such as hybridization-based approaches, reverse transcription quantitative PCR arrays (RT-qPCR), and NGS. Microarrays based on RNA-DNA hybrid capture are considered initial low-cost screening techniques, while NGS (gold standard technique) detects new miRNAs and different isoforms [84,85]. Nowadays, RT-qPCR is the most applicable technique, as it is considered a highly sensitive, specific, accessible, and reproducible tool.

However, there are many limitations and challenges related to antagomirs in human therapy. The first challenge is the deep characterization and analysis of miRNA sequences and targets, as the miRNAs may have multiple targets for which all functions and mechanisms of action should be well understood to be able to design the desired antagomirs using bioinformatics programs [86,87]. The second challenge is that the method of miRNA delivery for humans must be developed and optimized to avoid undesired tissue damage due to the high pressure of injection and electroporation by using alternative methods, such as viral vectors techniques [88,89,90]. For future clinical applications of miRNAs as biomarkers, standardized laboratory methods should be verified and more specific algorithms for data analysis and result interpretation should be established. Early diagnosis, disease progression, and treatment response prediction can be achieved through miRNAs’ development as biomarkers (Figure 3).

## 6. Conclusions and Future Prospects

In conclusion, many advancements have been made in our understanding of the way microRNAs work and how they can be used as biomarkers for different diseases. MicroRNAs, for example, can be useful for distinguishing patients with chronic HBV infections and non-infected patients. On the other hand, miRNAs change the expression of genes which are linked to liver disease caused by HCV. They also play an important role in the pathogenesis of HBV and HCV, a fact which represents the need for further research on advanced therapy options to fight infection with the hepatitis virus. Finally, it is expected that in the near future, the biological information about miRNAs that has been learned will be widely used in the medical field. Even though there are still many issues, miRNAs have an irrefutable potential for prognosis, diagnosis, and development of targeted therapy.

## Figures and Tables

**Figure 1 medicina-59-00173-f001:**
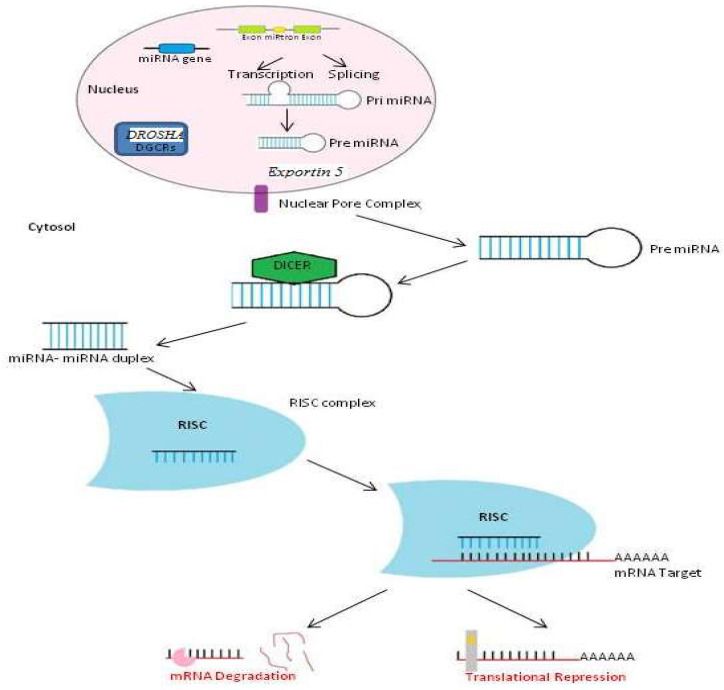
MicroRNAs’ biogenesis and mechanism of action. The miRNAs are formed from the pre-miRNA derived from pri-miRNA, which is transcribed from the genome by RNA polymerases (II and III) and forms mature microRNA after the generation of precursors with a cleavage event series. Pri-miRNA can be recognized and cut by DROSHA (nuclear ribonuclease III) to form a pre-miRNA that is exported from the nucleus to cytoplasm to be cleaved by the RNase III enzyme Dicer into a miRNA duplex of (18–22 nt). One strand of miRNA is usually incorporated into the RNA-induced silencing complex (RISC) to interact with its mRNA target. Finally, mature miRNA acts either by degrading the mRNA target or by inhibiting its translation.

**Figure 2 medicina-59-00173-f002:**
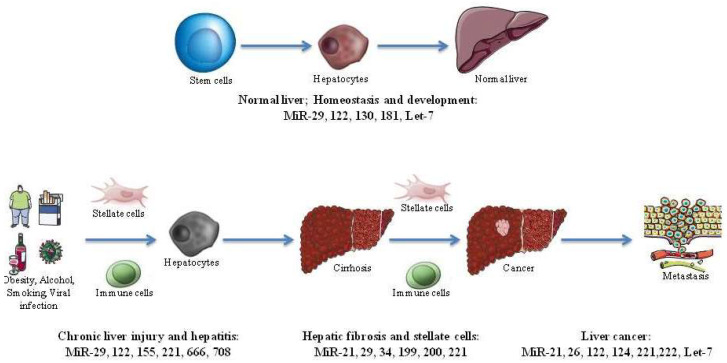
Important effects of microRNAs in chronic liver diseases and liver cancer. Summary of miRNAs functions identified in different liver disease such as chronic liver injury and hepatitis (MiR-29, 122, 155, 221, 666, and 708), hepatic fibrosis and stellate cells (Mir-21, 29, 34, 199, 200, and 221), and liver cancer (MiR-21, 26, 122, 124, 221, 222, and Let-7).

**Figure 3 medicina-59-00173-f003:**
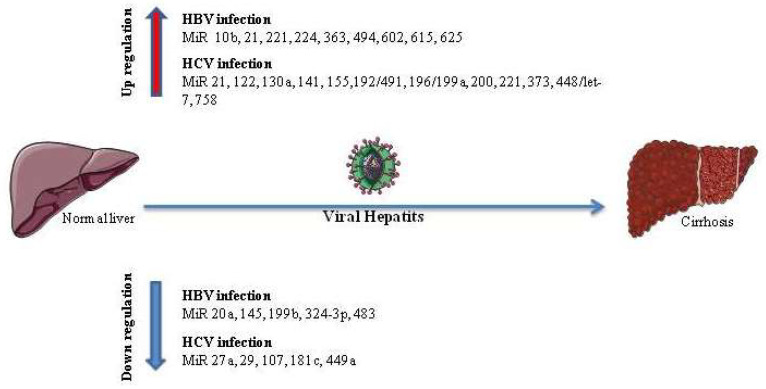
Up-regulated and down-regulated microRNAs in HBV and HCV infection. Altered microRNA expression (up-regulated and down-regulated) in HBV and HCV infection during the progression from viral hepatitis to liver cirrhosis.

## Data Availability

Data sharing is not applicable to this article as no new data were created or analyzed in this study.

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
