# Peer review of "MicroRNAs: Small Molecules with Significant Functions, Particularly in the Context of Viral Hepatitis B and C Infection"

_medicina, 2023, doi:10.3390/medicina59010173_

Round 1
Reviewer 1 Report
The title: Please write the word “Micrornas” as MicroRNAs”
Line 47: RISC abbreviation (RNA-induced silencing complex) is not included in abbreviation section.
Line 86: insert space between silencingcomplex.
Line 54: Please, rephrase this sentence “miRNAs are transcribed into pri-miRNAs, which are characterized….”. The miRNA are derived from the pri-miRNA not transcribed into pri-miRNAs.
Line 66: insert space between “ulatingmRNA”
Line 67: The phrase “There are two types of miRNAs: canonical and non-canonical” needs rephrasing to be “There are two pathways of miRNAs synthesis: canonical and non-canonical”.
Line 69: 30-UTR change to 3’UTR.
Line 71: The word “seed” what does it mean? Could it be “Deep”!
Line 72: 50 end change to 5’ end.
Figure 1 (out of the nucleus): The protein is called Exportin 5 NOT Exoprotin 5
Figure 1 (in the nucleus): The protein DROSHA not DRPSHA.
Figure 1 (out of the nucleus): Exportin5 works inside the nucleus. Please try to relocate the Exportin5 to be in the nucleus.
Line 81: When you mention a legend of a figure containing “RNA polymerases II and III”, you should find both enzymes in the illustration.
Line 81/82: The phrase “RNA polymerases II and IIIcan transcribed MicroRNAs into pri-miRNAs and formed” needs to be revised. The miRNAs are formed from the Pre-miRNA which are derived from Pri-miRNA which are transcribed from the genome by RNA polymerases.
Line 84: “to cleaved” change to “to be cleaved”
Line 85: “….Dicer in anmiRNA duplex”, I think it should be “….Dicer into an miRNA duplex”
Line 67-74: These phrases need comprehensive revision. The bypass one or more step or deviation from the classical pathway “canonical” is refered to be “NONCANONICAL pathway”. I think you may discuss using this reference (Biomol Concepts. 2014 Aug; 5(4): 275–287).
Figure 1: You have incorporated the mirtron (the first noncanonical pathway discovered), and also incorporated Drosha/Dgcrs. In the noncanonical pathway, both Drosha/Dgcrs are not needed. Therefore, you should differentiate either in text or by illustration to this point.
Line 116: “miRNAs roleas a biomarker research is still…”, rephrase to “miRNAs role in biomarker research is still….”
Several words are joined as in lines 87, 88, 95, 101, 111, 116, 126, 128, 203, 243, 249, 260, 278…etc, double commas in line 116,
Line 131: The legend of Figure 2 needs to be clarified. The different MiRNAs included in the figure should be indicated in the test of the legend, especially that the known miRNA is mentioned as MiR#.
Line 136: Delete “For instance,…”, and start the sentence directly by miR122.
Line 146: “NF-B pathway” change to “NF-kB pathway”
Lines 146/147: two successive “on the other hand”. Please rephrase.
Line 153/154: The sentence “-17-92 cluster (mir-17-5p, miR-18a, miR-19a, miR-19b, miR-20a, and miR-92a-1) in HCC, which is caused by HBV and miR21, in HBV-related HCC” is not understood. Please, try to clearly rephrase.
Line 156: “to,” delete the comma. In the same sentence, there are 6 commas that distract the meaning. The phrase needs revision. The same is repeated in the sentence in Lines 192-195. Again in Lines 220-225.
Line 188: “tumor-re-“, remove the hyphen between tumor and re.
Line 204: There is a number “36” without brackets.
Line 239: IRAK1 is not included in the abbreviation section.
Line 264: “….which shows a strong association”. The word “which” is not clear. Please rephrase to: “……are normalized to non-exosomal miRNA-16-5p. This normalization showed a strong association with the plasma of HCC patients”.
Line 267: “….markers; for example,”. Start a new sentence.
Line 274: “…has reported that”. Who has reported?
Line 280: “Liu et al.,[80]reported that ADAR1 plays an antiviral role in HBV infection” repeated words.
Figure 3: The legend of this figure needs to be more precise and clear. The figure does not show “Role of Micro RNAs in HBV and HCV replication and pathogenesis”. Please delete these words and keep the rest of the legend.
Line 318: “is linked..” change to “…are linked…”.
Reviewer 2 Report
The paper titled MicroRNAs: Small Molecules with Significant Functions, particularly in the context of viral hepatitis B and C infection gives us an overview of microRNAs and they role in liver diseases. The paper is nicely written, nice figures are made and used. Throughtout the paper there are a lot of joined words that should be separated to make the text easier to read. This paper provides us with some new information about the role of microRNAs in liver disease and use of microRNAs as potential biomarkers for liver disease.
